# Search for heavy scalar or pseudoscalar states in tt̄ events at CMS

**Laurids Jeppe** on behalf of the CMS collaboration

Deutsches Elektronen-Synchrotron DESY, Notkestr. 85, 22607 Hamburg, Germany

[laurids.jeppe@cern.ch](mailto:laurids.jeppe@cern.ch)

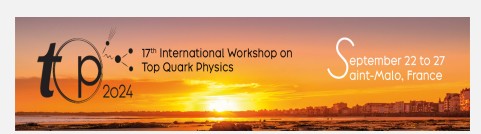

*The 17th International Workshop on*
*Top Quark Physics (TOP2024)*
*Saint-Malo, France, 22-27 September 2024*
doi:[10.21468/SciPostPhysProc.?](https://doi.org/10.21468/SciPostPhysProc.?)

## Abstract

A search for scalar or pseudoscalar states decaying to a top quark-antiquark pair (tt̄), using $138\,\mathrm{fb}^{-1}$ of pp collision data taken at $\sqrt{s} = 13\,\mathrm{TeV}$ using the CMS detector, is presented. Events with one or two leptons are analyzed using the invariant tt̄ mass ($m_{\mathrm{t\bar{t}}}$) as well as angular and spin correlation observables. An excess in the data is observed for low values of $m_{\mathrm{t\bar{t}}}$, preferring a pseudoscalar over a scalar hypothesis. It is interpreted in terms of a generic model of (pseudo)scalar boson production, as well as a simplified model of a tt̄ bound state ($\eta_{\mathrm{t}}$), yielding good agreement with the data. Moreover, limits on the couplings of additional (pseudo)scalar bosons to top quarks are set.

## 1 Introduction

The Higgs sector of the Standard Model (SM) is often considered a promising avenue to search for Beyond the Standard Model (BSM) physics, and many models including extended Higgs sectors, such as the Two-Higgs Doublet Model (2HDM) or supersymmetric models have been proposed. The new particles predicted in these models often exhibit Yukawa-like couplings to fermions, leading to large couplings to the top quark due to its large mass. In particular, for new electrically neutral states with masses above $2m_{\mathrm{t}}$, the decay to a top quark-antiquark pair (tt̄) is often dominant in large areas of parameter space, motivating searches for such states in tt̄ events at the LHC. In this work, heavy spin-0 bosons with scalar or pseudoscalar couplings to the top quark are considered.

At the same time, tt̄ bound states are expected to form in the SM according to several calculations (see e.g. Refs. [1–3]), manifesting as a broad peak in the invariant tt̄ mass spectrum slightly below the tt̄ threshold. At the LHC, they are expected to be dominated by a pseudoscalar component, leading to a similar signature as an additional pseudoscalar boson, which suggests a search for these effects using the same methodology.

This work presents such a search [4], performed with the CMS experiment [5] in proton-proton collisions at the LHC with the full Run 2 dataset, corresponding to an integrated luminosity of $138\,\mathrm{fb}^{-1}$. A similar search was presented by ATLAS in Ref. [6]. The search presented

here updates on a previous work, which considered only $35.9\,\mathrm{fb}^{-1}$ of data [7]. Two analysis channels are considered, targeting the dilepton ($\ell\ell$) and lepton+jets ($\ell$j) decay channels of t$\bar{\mathrm{t}}$, and the invariant t$\bar{\mathrm{t}}$ mass as well as angular and spin correlation observables are employed to isolate the signals from the SM background.

## 2   Signal modeling

For the interpretation of a generic pseudoscalar or scalar boson (denoted A and H, respectively), the signal is assumed to be produced in gluon fusion through a top quark loop, with only a Yukawa-like coupling to the top quark considered. As a result, the free paremeters of the model are the masses, widths and coupling strength modfiers ($g_{\mathrm{At\bar{t}}}$ resp. $g_{\mathrm{Ht\bar{t}}}$). Because the resulting final state is the same as in t$\bar{\mathrm{t}}$ production in the SM, interference with the SM is expected, leading to a peak-dip structure in the $m_{\mathrm{t\bar{t}}}$ spectrum.

For t$\bar{\mathrm{t}}$ bound states, on the other hand, no full calculation that can be directly compared to data is available at the time of writing. Calculations of the expected $m_{\mathrm{t\bar{t}}}$ spectrum can be performed in the framework of non-relativistic QCD (NRQCD) [1], which predicts an attractive potential and a resulting peak for the color-singlet component of t$\bar{\mathrm{t}}$, and a repulsive potential, leading to a suppresion, for the color-octet component. In this work, a simplified model (proposed in Ref. [8]) for this peak is used, denoted $\eta_{\mathrm{t}}$. It consists of a generic spin-0, color-singlet, pseudoscalar resonance coupling directly to gluons and top quarks, with the mass and width of the resonance extracted from a fit to an NRQCD prediction, which yields $m_{\eta_{\mathrm{t}}} = 343\,\mathrm{GeV}$. To not influence the t$\bar{\mathrm{t}}$ continuum, the model is restricted to invariant masses of $|m_{\mathrm{t\bar{t}}} - m_{\eta_{\mathrm{t}}}| < 6\,\mathrm{GeV}$. The prediction from this model is then added to the continuum t$\bar{\mathrm{t}}$ prediction as given by perturbative QCD (pQCD). While this model is not expected to reproduce the details of the predicted lineshape, which is anyway not well known in the first place, it is deemed sufficient for this analysis due to the coarse experimental $m_{\mathrm{t\bar{t}}}$ resolution rendering the details irrelevant.

## 3   Analysis setup

In the $\ell$j channels, events with exactly one lepton (e or $\mu$) and 3 or more jets, of which at least 2 are b-tagged, are selected and sorted into four categories based on the lepton flavor as well as the number of jets (3 or $\geq$4). The NeutrinoSolver algorithm [9] is used to reconstruct the t$\bar{\mathrm{t}}$ system, and an energy correction factor is applied for events with exactly three jets to account for jets that were merged or lost. Two-dimensional templates are constructed based on $m_{\mathrm{t\bar{t}}}$ and $|\cos\theta^{\star}|$, where $\theta^{\star}$ is the scattering angle of the leptonically decaying top quark with respect to the beam axis. This variable has discriminating power because SM t$\bar{\mathrm{t}}$ production peaks at small scattering angles, while the A, H and $\eta_{\mathrm{t}}$ signals are isotropic.

In the $\ell\ell$ channels, events with exactly two leptons and at least 2 jets, with at least one b-tagged, are selected. They are split into categories by lepton flavor, and in the same-flavor channels additional cuts are applied to reject Z+jets events. Again, the four-momenta of the t$\bar{\mathrm{t}}$ system is reconstructed, employing an analytical approach which assumes that the two neutrinos in the t$\bar{\mathrm{t}}$ decay are the sole source of missing transverse momentum ($p_{\mathrm{T}}^{\mathrm{miss}}$) and that the top quarks and W bosons are on-shell. The finite detector resolution is taken into account by repeating the reconstruction 100 times per events with inputs (lepton and jet four-momenta as well as $p_{\mathrm{T}}^{\mathrm{miss}}$) randomly smeared and taking a weighted average over all real solutions. Three-dimensional templates are constructed from $m_{\mathrm{t\bar{t}}}$ as well as two spin correlation observables $c_{\mathrm{hel}}$ and $c_{\mathrm{han}}$. The distributions of both of these observables (before phase space cuts) are straight

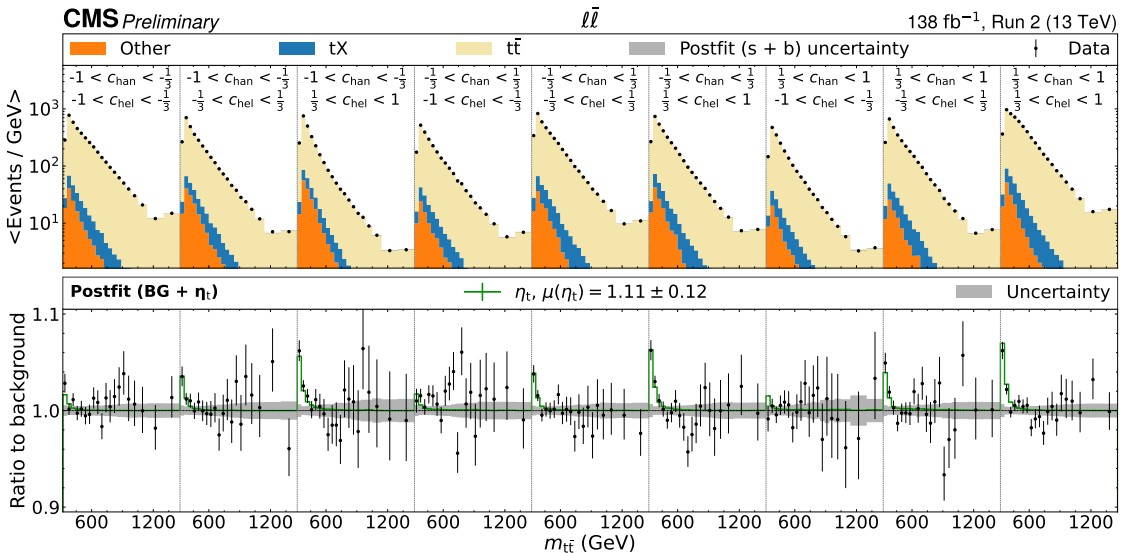

Figure 1: Postfit three-dimensional distribution of $m_{t\bar{t}}$, $c_{hel}$ and $c_{han}$ in the $\ell\ell$ channel in a signal+background fit with $\eta_t$ as the signal. The perturbative QCD SM predictions are shown in the upper panel as the colored stack and the observed data as the black dots. The lower panel shows the ratio of data to the pQCD prediction, the postfit uncertainty (grey band), and the $\eta_t$ signal normalized to the postfit yield (green line). This figure is taken from the auxiliary material of Ref. [4].

lines, whose slopes depend on entries in the $t\bar{t}$ spin density matrix and thus directly probe the spin state of the $t\bar{t}$ system. For pure $^1S_0$ states, as produced by the pseudoscalars A and $\eta_t$, the slope of $c_{hel}$ is maximally positive, while for pure $^3P_0$ states (produced by the scalar H), the slope of $c_{han}$ is maximally negative. As a result, the observables provide good discriminations for the different signal hypotheses.

The dominant background in all channels consists of SM $t\bar{t}$ production, which is generated at NLO in QCD using Powheg v2, interfaced to Pythia 8, and reweighted to higher order predictions at NNLO in QCD and NLO in electroweak processes using a two-dimensional bins of $m_{t\bar{t}}$ and the top scattering angle $\cos\theta^\star$. Furhter backgrounds are tW and t-channel single-top production (estimated from MC), Z+jets production ($\ell\ell$ only, estimated from MC with data-driven normalization) and QCD multijet as well as W+jets production ($\ell$j only, estimated from a data sideband).

## 4 Results

Across all channels, an excess of data compared to the continuum pQCD prediction is observed at low values of $m_{t\bar{t}}$, translating directly to an excess of the observed over the expected limits for the coupling strengths $g_{At\bar{t}}$ and $g_{Ht\bar{t}}$. The excess is found to be stronger in the pseudoscalar case A.

Based on these observations, the excess is interpreted as a possible $t\bar{t}$ bound state using the $\eta_t$ model as described in Sec. 2 by extracting the $\eta_t$ cross section, corresponding to the difference of the observed data yield to the pQCD prediction, in a signal+background fit.

The resulting cross section is $\sigma(\eta_t) = 7.1 \pm 0.8$ pb. By comparing to the prediction given in Ref. [8] of $\sigma(\eta_t)^{pred} = 6.43$ pb, obtained by fitting a NRQCD prediction, good agreement is found. The parametrized model seems to describe the data well, as shown as an example for the $\ell\ell$ channel in form of a postfit distribution in Fig. 1. It can in particular be seen that

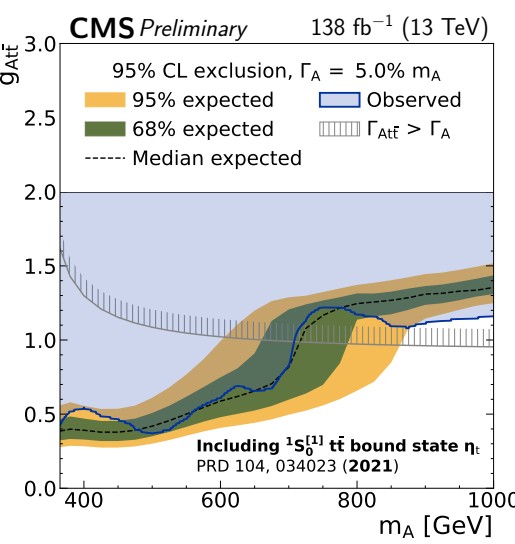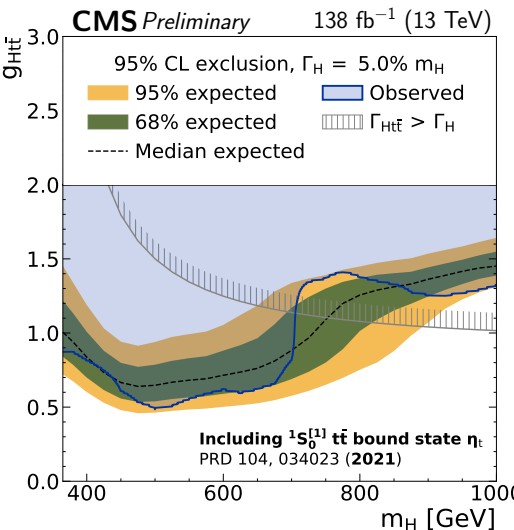

Figure 2: Exclusion limits at 95% confidence level on the coupling strength modifier of the pseudoscalar A (left) and scalar H (right) as a function of their respective mass for an example width of 5%. The observed (expected) limit is shown in blue (black), and the 68% (95%) intervals are shown as the green (yellow) band. This figure is taken from Ref. [4].

both the signal simulation and the data exhibit an increase in the slope of $c_{hel}$ in the low $m_{t\bar{t}}$ bins compared to the pQCD prediction, which is consistent with a pseudoscalar contribution. Nonetheless, it should be cautioned that the $\eta_t$ model considered here is not a complete description of a $t\bar{t}$ bound state, but a simple parametrization, missing for example (but not limited to) contributions from soft gluons changing color-octet into color-singlet states.

The uncertainty on the $\eta_t$ cross section is dominated by systematic effects from the pQCD background modeling, including the electroweak corrections to $t\bar{t}$, parton shower uncertainties, missing higher orders in the matrix elements, PDF uncertainties and the uncertainty in the top quark mass. For a full overview of the systematic uncertainties, see Ref. [4].

Following the observation that the data is well described by the $\eta_t$ model, limits on additional (BSM) spin-0 states A and H are set by including the $\eta_t$ contribution in the background prediction, with its normalization freely floating in the fit. The excess is now no longer present, and limits on the couplings $g_{At\bar{t}}$ and $g_{Ht\bar{t}}$ are set in the mass range of $365 - 1000$ GeV for different A/H widths. An example is shown in Fig. 2 for a relative width of 5%.

## 5   Conclusion

A search for heavy scalar or pseudoscalar spin-0 states in $t\bar{t}$ events, performed with the CMS detector with $138\,\text{fb}^{-1}$ of proton-proton collision data at $\sqrt{s} = 13$ TeV, is presented [4]. The dilepton and lepton+jets decay channels of $t\bar{t}$ are analyzed using the invariant $t\bar{t}$ mass as well as angular and spin correlation observables. An excess, localized in low bins of $m_{t\bar{t}}$, is observed, and found to be consistent with a pseudoscalar state. It is interpreted in terms of a generic model for scalar or pseudoscalar boson production (A or H), as well as a parametrized model of a $t\bar{t}$ bound state ($\eta_t$). The cross section of $\eta_t$, understood as the difference to the pQCD background prediction, is extracted as $\sigma(\eta_t) = 7.1 \pm 0.8$ pb. Good agreement with the data is found, and stringent exclusion limits are set on the couplings of A and H by adding $\eta_t$ to the background prediction.

**Funding information**    L.J. acknowledges support by the Deutsche Forschungsgemeinschaft (DFG, German Research Foundation) under Germany's Excellence Strategy – EXC 2121 "Quantum Universe" – 390833306. This work has been partially funded by the Deutsche Forschungsgemeinschaft (DFG, German Research Foundation) - 491245950.

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
