# Peer review of "Search for heavy scalar or pseudoscalar states in $\mathrm{t \bar{t}}$ events at CMS"

_SciPost Physics Proceedings_

## Round 1 · Referee Report · Samuel Calvet (Referee 1) · 2025-1-7

Strengths

1- Important result revealing potential new bound state, the first one in the top quark sector
2- New limits on the A/H production with the largest dataset analyzed by CMS for this search (138fb-1)

Weaknesses

1- Some sentences need to be clarified
2- Some important details could be added

Report

This proceedings are a very good summary of the very interesting result presented at TOP2024, on the search for new (pseudo)scalar decaying into top quark pairs.
It reports the evidence of a new bound state, ηtop, the very first one in the top sector. Assuming this new state, predicted by the Standard Model of particle physics, it provided new limits on the production of Beyond the Standard Model (pseudo)scalars.
I highly recommend its publication once a few minor changes have been made.

Requested changes

1- " It consists of a generic spin0, color-singlet, pseudoscalar resonance coupling directly to gluons and top quarks". Could you provide the couplings values that have been used ? And what is the actual width of ηtop
2- " due to the coarse experimental mt¯t resolution ". Could you provide the values of the resolution ? This detail was provided in the answer to questions asked at Saint-Malo
3- "an energy correction factor is applied for events with exactly three jets" Could you say on what assumptions this factor is computed ?
4- Explain what are chel and chan or provide a reference

Recommendation

Ask for minor revision

---

## Round 2 · Author Response

Thank you for your report and your recommendation to publish the proceedings. Please find below my answers to your comments.

  1. The couplings of the toy model are arbitrary and irrelevant since they only determine the cross section, which is the free-floating POI in the fit. The width is set to 7 GeV in the PAS presented at Saint-Malo, which I added to the text.
  2. The resolution is ~15% at the ttbar threshold, which I added to the text.
  3. This method is described in https://arxiv.org/abs/1310.3263, which I’ve added as a reference. It roughly works as follows: It is assumed that the lost or merged jet is one of the light jets from the hadronic W decay. A proxy for the hadronic top quark is defined as the vectorial sum of one b jet and one light jet, with a likelihood criterion used for the assignment. The four-momentum of this proxy is then rescaled as a function of its invariant mass, such that the peak of the reconstructed mtt distribution in simulation coincides with the generator-level peak. The same function is used for the rescaling in simulation and data. Since the procedure is somewhat complicated, and the lepton+3jets channel the least sensitive, I would prefer not to explain it in detail in the text.
  4. I’ve added a short explanation as well as two references.

---

## Round 2 · List of Changes

• Added the width of 7 GeV for the EtaT toy model considered.
  • Added the mtt resolution of ~15%.
  • Added a reference to https://arxiv.org/abs/1310.3263 regarding the energy correction in the lepton+3jets channels.
  • Added a description of the spin correlation observables chel and chan: "Three-dimensional templates are constructed from $\mtt$ as well as two spin correlation observables $\chel$ and $\chan$. They are defined in the helicity basis for the top and antitop spins, corresponding to boosting the lepton four-momenta into the rest frames of their parent top quarks. The observable $\chel$ is the scalar product of the lepton directions of flight in the helicity basis, while $\chan$ is defined as a similar scalar product containing a negative sign in the top quark direction of flight. "
  • Added two references for chel and chan and the helicity basis: https://arxiv.org/abs/2401.08751 and https://arxiv.org/abs/1508.05271.
  • Fixed several spelling mistakes.

---

## Editorial Decision

accepted_in_target_journal